# Actionable absolute risk prediction of atherosclerotic cardiovascular disease based on the UK Biobank

**Ajay Kesar** [ID]**[1]** *, **Adel Baluch[1], Omer Barber[1], Henry Hoffmann[1], Milan Jovanovic[1], Daniel Renz[1], Bernard Leon Stopak[1], Paul Wicks[1], Stephen Gilbert[1,2]**

**1** Ada Health GmbH, Berlin, Germany, **2** EKFZ for Digital Health, University Hospital Carl Gustav Carus Dresden, Technische Universität Dresden, Dresden, Germany

* science@ada.com

## Abstract

Cardiovascular diseases (CVDs) are the primary cause of all death globally. Timely and accurate identification of people at risk of developing an atherosclerotic CVD and its sequelae is a central pillar of preventive cardiology. One widely used approach is risk prediction models; however, currently available models consider only a limited set of risk factors and outcomes, yield no actionable advice to individuals based on their holistic medical state and lifestyle, are often not interpretable, were built with small cohort sizes or are based on lifestyle data from the 1960s, e.g. the Framingham model. The risk of developing atherosclerotic CVDs is heavily lifestyle dependent, potentially making many occurrences preventable. Providing actionable and accurate risk prediction tools to the public could assist in atherosclerotic CVD prevention. Accordingly, we developed a benchmarking pipeline to find the best set of data preprocessing and algorithms to predict absolute 10-year atherosclerotic CVD risk. Based on the data of 464,547 UK Biobank participants without atherosclerotic CVD at baseline, we used a comprehensive set of 203 consolidated risk factors associated with atherosclerosis and its sequelae (e.g. heart failure). Our two best performing absolute atherosclerotic risk prediction models provided higher performance, (AUROC: 0.7573, 95% CI: 0.755–0.7595) and (AUROC: 0.7544, 95% CI: 0.7522–0.7567), than Framingham (AUROC: 0.680, 95% CI: 0.6775–0.6824) and QRisk3 (AUROC: 0.725, 95% CI: 0.7226–0.7273). Using a subset of 25 risk factors identified with feature selection, our reduced model achieves similar performance (AUROC 0.7415, 95% CI: 0.7392–0.7438) while being less complex. Further, it is interpretable, actionable and highly generalizable. The model could be incorporated into clinical practice and might allow continuous personalized predictions with automated intervention suggestions.

## Introduction

Globally, cardiovascular diseases (CVDs) are the number one cause of all death [1, 2]. In 2016, 17.9 million people died of CVDs alone, accounting for 31% of all global deaths [1]. The direct

**Data Availability Statement:** There are restrictions prohibiting the direct provision of UK Biobank data used in this manuscript. The data were obtained from a third party, UK Biobank, upon application

under accession/application number 34802. Interested parties can apply for the data from UK Biobank directly, at http://www.ukbiobank.ac.uk. The UK Biobank will consider data applications from bona fide researchers for health-related research that is in the public interest. This process for reader access to the UK Biobank is the same process as followed by the authors of this study and will provide the same data. All extracted columns are stated in the supporting information files S1 Table and S2 Table.

**Funding:** This research was funded by Ada Health GmbH and has been conducted using the UK Biobank under application id 34802.

**Competing interests:** This research was funded by Ada Health GmbH and has been conducted using the UK Biobank under application number 34802. All of the authors are or were employees of, contractors for, or hold equity in Ada Health GmbH. AK, AB, OB, HH, MJ, DN, BLS and SG are employees or company directors of Ada Health GmbH and some of the listed authors hold stock options in the company. Ada Health GmbH has received research grant funding from the Bill & Melinda Gates Foundation, Fondation Botnar, the Federal Ministry of Education and Research Germany, the Federal Ministry for Economic Affairs and Energy Germany and the European Union. PW is employed by Wicks Digital Health Ltd, which has received funding from Ada Health, AstraZeneca, Baillie Gifford, Biogen, Bold Health, Camoni, Compass Pathways, Coronna, EIT, Endava, Happify, HealthUnlocked, Inbeeo, Kheiron Medical, Lindus Health, Sano Genetics, Self Care Catalysts, The Learning Corp, The Wellcome Trust, THREAD Research, VeraSci, and Woebot. HH is the topic driver of the AI-based symptom assessment group of the WHO/ITU Focus Group on AI4H (Artificial Intelligence for Health) and SG is a member of the clinical evaluation topic group of the WHO/ITU Focus Group on AI4H. A related patent application is currently pending with the title "System and method for predicting the risk of a patient to develop an atherosclerotic cardiovascular disease" and application number EP21191089.8. This does not alter our adherence to PLOS ONE policies on sharing data and materials.

costs of CVDs in the US for 2010 were $272.5b whereas indirect costs were $171.7b and are expected to increase to $818.1b and $275.8b in 2030 respectively [3, 4]. Atherosclerosis alone is responsible for 1.3% of all hospital stays with costs of $9b per year, while all atherosclerosis-related diseases amount to $43.5b of hospital costs annually [5]. Individually, patients with CVD incur more than twice the medical costs of age- and sex-matched patients without CVD, largely because of the increased likelihood of subsequent hospitalizations. The greatest differences in total CVD costs usually occur when comparing patients with and without a secondary CVD hospitalization [6].

All current guidelines on the prevention of CVD in clinical practice recommend the assessment of total CVD risk since atherosclerosis is usually the product of a number of risk factors [7, 8] and in recent years these guidelines have evolved to focus on the absolute risk of disease as opposed to relative risk [7–10]. Clinician tools for CVD risk estimation must enable rapid and accurate estimation of an individual patient's absolute CVD risk [7], or for opportunistic screening of high-risk patients from relevant populations [11]. Screening is the identification of unrecognized disease or risk of disease in individuals without symptoms. In addition to opportunistic screening, which is carried out without a predefined strategy (e.g. when the individual is consulting a general practitioner (GP) for some other reason), tools can be used for systematic screening, which is centrally organized strategic screening in the general population or in targeted subpopulations, such as subjects with a family history of premature CVD or familial hyperlipidaemia [7]. There is ongoing debate on the role of systematic centralized population based screening in CVD [10, 12] because of burdensome diagnostic testing following the use of risk based screening tools [13]. A relatively new area of screening is self-screening, carried out by proactive individuals using screening tools on mobile devices such as smartphones or smartwatches, which may use built in app-linked sensors or screening chat-bots [14–16]. There is public demand for reliable, actionable, explainable and usable health information tools [17], including for disease screening.

The risk to build up atherosclerotic plaque varies and is determined by multiple factors such as genetics, environment and lifestyle [11, 18–21]. The risk of developing atherosclerotic plaque can be reduced based on an individual's behavioral risk factors, such as smoking, physical activity and nutrition [1, 11, 19, 20].

Most diseases, including atherosclerotic CVDs, have a complex pathophysiology that involves multiple interacting molecular systems, making it insufficient to look only at an isolated biological pathway or a subset of markers to predict disease risk [22]. A precision medicine based approach is required, where multiple biological layers are considered (i.e., 'multiomics'), alongside clinical and lifestyle data [22]. Such an approach has the potential to capture all important interactions or correlations detected between molecules in different biological layers, providing a holistic understanding of an individual's current health status and enabling the quantification of an individual's absolute risk of atherosclerotic CVDs [23, 24].

Previous studies in this area use a limited set of risk factors and outcomes for their analyses [7, 25, 26]. In recent years, the knowledge of behavioral risk factors and of the pathophysiology of atherosclerotic CVDs have advanced tremendously [11, 25]. Current absolute risk prediction models have limited predictive capability as they have not been trained on all possible atherosclerotic CVD outcomes [27–29], or they include outcomes which are unmodifiable such as those related to pregnancy, accidents, or congenital factors [29].

Both SCORE (Systematic COronary Risk Evaluation) and SCORE2 [30, 31], are models for predicting relative CVD risk, whereas we focus on predicting absolute CVD risk, which is why we chose to omit those models from our analysis. Another related investigation, which also used the UK Biobank (UKB) dataset, developed multiple Cox Proportional Hazard models for 10-year CVD risk prediction, with a reduced version requiring 47 risk factors and another

version disregarding all cholesterol risk factors as well as systolic blood pressure, in order to provide a simple approach for risk prediction in remote settings with limited testing resources [32]. However, survival models such as the proportional hazard model are not designed to provide absolute risk estimates for individual patients.

Machine learning (ML) based approaches have many advantages compared to humans or standard statistical algorithms, such as superior performance, being able to identify complex non-linear patterns, the ability to encode diverse and high dimensional data types, being more stable to outliers, allowing continuous model updates, versatility for different domains and scalability [33–36].

However, classic disadvantages of ML based approaches are their lack of interpretability, risk for inherent bias due to the used data, difficulty to acquire physician adoption, explaining to physicians why a new risk model might be superior to existing ones, with all of these hindering widespread adoption of ML based risk prediction models [36, 37]. One example for ML based CVD risk prediction is the AutoPrognosis based approach, where an ensemble of multiple ML pipelines has also been applied on the UK Biobank dataset for 5-year CVD risk prediction [29]. Further, using a purely ML-driven approach can lead to a model that requires too many risk factors to compute risk, which is infeasible for routine clinical check-ups. Another disadvantage of purely data-driven approaches is the inclusion of risk factors which might show strong correlations but are unrelated to the pathophysiology of CVDs or are not actionable, making them inapplicable in a clinical setting or as an actionable self-management tool [29].

The aim of this study was to use a large-data ML approach to develop an actionable absolute risk prediction tool which considers the holistic health of an individual. Uniquely, we focused on behavioral risk factors relating to all atherosclerotic CVD outcomes. Our goal was to have a holistic understanding of an individual's current health status, to better quantify their risk of atherosclerotic CVDs, and to provide actionable advice. Our approach is novel in that we employ a highly holistic understanding of an individual's current health status, to better quantify their risk of all atherosclerotic CVDs. By utilizing a comprehensive set of lifestyle factors, we enable the subsequent suggestion of personalized and actionable advice relating to unhealthy risk factors. Instead of using only a limited set of risk factors, we aimed to achieve this by taking multiple biological layers into account, which include: (i) multi-omics data from blood samples (e.g. lipidome and proteome); (ii) family history (e.g. genome), (iii) lifestyle data, (iv) clinical data and (v) environmental data; along with (vi) an extensive set of risk factors and outcomes.

We used data from 464,547 participants of the UK Biobank study who did not have atherosclerotic CVD at their baseline visit. We created an automated pipeline to benchmark risk prediction classifier algorithms against each other, then evaluated their predictive performances in the overall population and tested the generalizability of the top-performing classifiers through retraining and testing on different sub-populations. We explored the clinical implications of the proposed classifiers, with a focus on the top-performing models. This study does not focus on the algorithmic aspects of the utilized classifiers.

Methodological details on the utilized classifiers can be found in the open-source documentation of the respective algorithms of the scikit-learn [38] and xgboost [39] libraries and in the supporting information (S4 Table).

## Materials and methods

Baseline data from the UK Biobank was utilized to extract an extensive set of risk factors and outcomes associated with the pathophysiology of atherosclerotic CVDs. A benchmarking pipeline was used to train and evaluate different standard and ML algorithms for the task of 10-year atherosclerotic CVD risk prediction. The performance was measured using AUROC

and compared against the baseline models Framingham and QRisk3, which are widely used and recommended models. We evaluated our best performing models further by analyzing the most informative features and assessed model generalizability and created a reduced model.

## Study design and participants

The UK Biobank is a long-term prospective large-scale biomedical database including over 500,000 participants aged 40–69 years (when recruited between 2006 and 2010). The database is globally accessible to approved researchers undertaking research into the most common and life-threatening diseases and continuously collects phenotypic and genotypic data about its participants, including data from questionnaires, physical measures, blood, urine and saliva samples, lifestyle data [40]. This data is further linked to each participant's health-related records, accelerometry, multimodal imaging, genome-wide genotyping and longitudinal follow-up data for a wide range of health-related outcomes [40, 41]. The UK Biobank study protocol is available online [42].

The North West Multi-Centre Research Ethics Committee approved the UK Biobank study and all participants provided written informed consent prior to study enrollment. Our research is covered by the UK Biobank's Generic Research Tissue Bank (RTB) Approval and was approved by the UK Biobank Access Management Team [43].

We excluded participants with atherosclerotic CVDs present before or during baseline, participants who chose to leave the UKB study and participants who were lost due to various reasons. The resulting cohort consisted of 464,547 participants. The last available date of participant follow-up was March 5th, 2020.

**Risk factor definition.** We curated a list of all generally known risk factors and outcomes for atherosclerotic CVDs from the medical literature and from validated risk prediction models. This preliminary list of risk factors was reduced through curation to focus on those factors that were clearly involved in the pathophysiology of atherosclerosis and those that are modifiable through behavioral change. The curation was carried out by three medical doctors with experience in diagnosing or scientifically modelling cardiovascular diseases. We consolidated all relevant UKB columns into 203 risk factors and grouped them into six categories: demographics (e.g. age, biological sex, ethnicity), biomarkers (e.g. cholesterol, glucose, blood pressure, heart rate), lifestyle (e.g. alcohol consumption, smoking, physical activity, sleep, social visits), environment (e.g. exposure to tobacco smoke, work and housing and other socio-economic related factors), genetics (e.g. family history of CVD, stroke, diabetes, high cholesterol, high blood pressure) and comorbidities (e.g. heart arrhythmias, diabetes, acute & chronic kidney injury, migraines, rheumatoid arthritis, systemic lupus erythematosus, severe mental illnesses (schizophrenia, bipolar disorder, depression, psychosis), diagnosis or treatment of erectile dysfunction, atypical antipsychotic medication). A categorized list of all risk factors used in our analysis is provided in the supplementary data (S1 Table).

**Outcome definition.** In the same manner as described above, an initial list of atherosclerotic CVDs was further reviewed and curated by the same team of medical doctors. All resulting CVDs of interest are associated with atherosclerotic plaque build-up, are modifiable and relate to the collected risk factors only. Thus, we disregard brain haemorrhages due to accidents and congenital and pregnancy-related CVDs, which are not actionable. The curated list of all ICD-10 and ICD-9 outcomes meeting the above criteria consists of 193 total (125 unique) CVD outcomes, e.g. coronary/ischaemic heart disease, heart attack, angina, stroke, cardiac arrest, congestive heart failure, left ventricular failure, myocardial infarction, aortic valve stenosis, cerebral artery occlusions, nontraumatic haemorrhages. A list with all outcome codes used in our analysis is provided in the supplementary data (S2 Table). An atherosclerotic CVD

event was defined as the first occurrence out of the following: any of the atherosclerotic CVD outcome diagnosis codes, also as primary or secondary death cause during the 10-year follow-up period.

**Cohort follow-up.** Follow-up time was set to 10 years as commonly used in other risk models (see Table 2 in [7]) and counted from the date of initial assessment center visit. Individuals who died from other causes during their follow-up period or had a relevant CVD event past their individual follow-up period, were marked as not having had a relevant CVD event.

## Models used in comparison

**Framingham risk score.** The Framingham 10-year CVD absolute risk score is based on the data of the two prospective studies, the Framingham Heart Study and the Framingham offspring study [27]. The cohort consists of 8491 participants, with 4522 women and 3969 men who attended a baseline examination between 30 and 74 years of age and were free of CVD. A positive CVD outcome was defined as any of the following: coronary death, myocardial infarction, coronary insufficiency, angina, ischemic stroke, hemorrhagic stroke, transient ischemic attack, peripheral artery disease and heart failure.

Participants were followed up for 12 years where 1174 participants developed a CVD. Two biological sex-specific risk models were derived, with one model using lipid measurements and the other one Body Mass Index (BMI). The variables used were biological sex, age, total cholesterol, HDL cholesterol, treated and untreated systolic blood pressure, smoking status and diabetes status.

The Framingham risk calculators and model coefficients are publicly available [44]. We imputed missing data using simple mean imputation.

**QRisk3.** The QRisk3 10-year CVD absolute risk score is based on a prospective open cohort study using data from general practices (GPs), mortality and hospital records in England [28]. The cohort consists of 10.56 million patients between the age of 25 and 84 years, where 75% of the patients were used for training and 25% for validation. Patients with a pre-existing CVD, missing Townsend score or using statins were removed from the baseline. Patients were classified as having a positive CVD outcome when any of the following outcomes was present during follow-up in the GP, hospital or mortality records: coronary heart disease, ischaemic stroke, or transient ischaemic attack. QRisk3 used the following ICD-10 codes: G45 (transient ischaemic attack and related syndromes), I20 (angina pectoris), I21 (acute myocardial infarction), I22 (subsequent myocardial infarction), I23 (complications after myocardial infarction), I24 (other acute ischaemic heart disease), I25 (chronic ischaemic heart disease), I63 (cerebral infarction), and I64 (stroke not specified as haemorrhage or infarction). The utilized ICD-9 codes were: 410, 411, 412, 413, 414, 434, and 436. Participants were followed-up for 15 years where 363,565 participants of the training set (4,6%) developed a relevant CVD. One biological sex-specific risk model was derived.

The risk factors used in the final model were age, ethnicity, deprivation, systolic blood pressure, BMI, total cholesterol/HDL cholesterol ratio, smoking status, family history of coronary heart disease, diabetes status, treated hypertension, rheumatoid arthritis, atrial fibrillation, chronic kidney disease, systolic blood pressure variability, diagnosis of migraine, corticosteroid use, systemic lupus erythematosus, atypical antipsychotic use, diagnosis of severe mental illnesses, diagnosis or treatment of erectile dysfunction.

The QRisk3 risk calculator and model coefficients are publicly available [45], built into all major NHS GP systems and included in the UK's national guidelines (https://www. healthcheck.nhs.uk/seecmsfile/?id=1687, accessed 10th November 2021). We imputed missing data using simple mean imputation.

**Standard linear and ML models.** Since the introduction of the classic CVD risk prediction methods, the field of supervised machine learning has developed from classical statistics with the sole purpose of maximizing predictive accuracy with modern statistical methods. Therefore, in addition to using standard linear models, we tested the major ML approaches, covering a wide spectrum of the possible ML design space, to evaluate which model type performs best for our task. Based on our initial benchmarking pipeline results, we focused on reporting the results of the initially best performing models: logistic regression, random forest and XGBoost.

We compared regularized linear regression (with L1 penalty), random forests and gradient boosting (xgboost implementation) for assessing the highest achievable Area Under the Receiver Operating Characteristic Curve (AUROC) value, which we used for assessing the trade-off between number of features and predictive performance of several simpler *practical risk predictors*, as determined by an iterative feature elimination procedure outlined below. L1 regularization for logistic regression implements a strong penalty for non-zero feature weights, resulting in a feature selection procedure that discards features that are likely to be non-predictive. Random Forest is an ensemble method that fits many decision trees independently to a subset of the data. We implemented both methods using their scikit-learn library implementation. Finally, we evaluated Extreme Gradient Boosting: Gradient boosting is an ensemble tree-based machine learning method that combines many weak classifiers to produce a stronger one. It sequentially fits a series of classification or regression trees, with each tree created to predict the outcomes misclassified by the previous tree [46]. By sequentially predicting residuals of previous trees, the gradient boosting process has a focus on predicting more difficult cases and correcting its own shortcomings. Extreme Gradient Boosting (XGB / XGBoost) is a specific implementation of the gradient boosting process, and uses memory-efficient algorithms to improve computational speed and model performance [39, 47].

For completeness, we briefly evaluated a number of other standard classifiers, but discarded them due to excessive computational complexity or inferior performance so we do not report their performances here: Decision Trees [48], Voting Classifiers, Multi-Layer Perceptrons with 2 layers and 200 and 150 neurons each (Neural Network) [49], stochastic gradient descent implementing a support vector machine algorithm [50, 51], Ada Boost [52, 53], Gradient Boosting [46], K Neighbors [54], Quadratic Discriminant Analysis [55] and Gaussian Naive Bayes [38, 56].

## Model development and benchmarking using pipeline

We built a benchmarking pipeline for automated and reproducible data extraction, normalization, imputation, model training, tuning of model hyperparameters, classification, documentation and reporting.

We implemented all models using their respective scikit-learn library or xgboost library implementation using the Python programming language [38, 39]. Details on the used Python libraries, methods and parameters are provided in the supplementary data (S3 and S4 Tables).

Categorical values were one-hot encoded. Data normalization was performed by removing the mean and scaling to unit variance. Data imputation was performed for all models using a simple mean imputation. The models' hyper-parameters were determined using grid search and stratified k-fold cross validation using 3 folds was employed to avoid overfitting.

Finally, we assessed model performance mainly using the AUROC. Fig 1 visualizes an overview of all performed steps of our experimental setup.

**Iterative feature elimination.** We employed an iterative feature elimination procedure based on the regularized logistic regression for finding the best trade-off between predictive

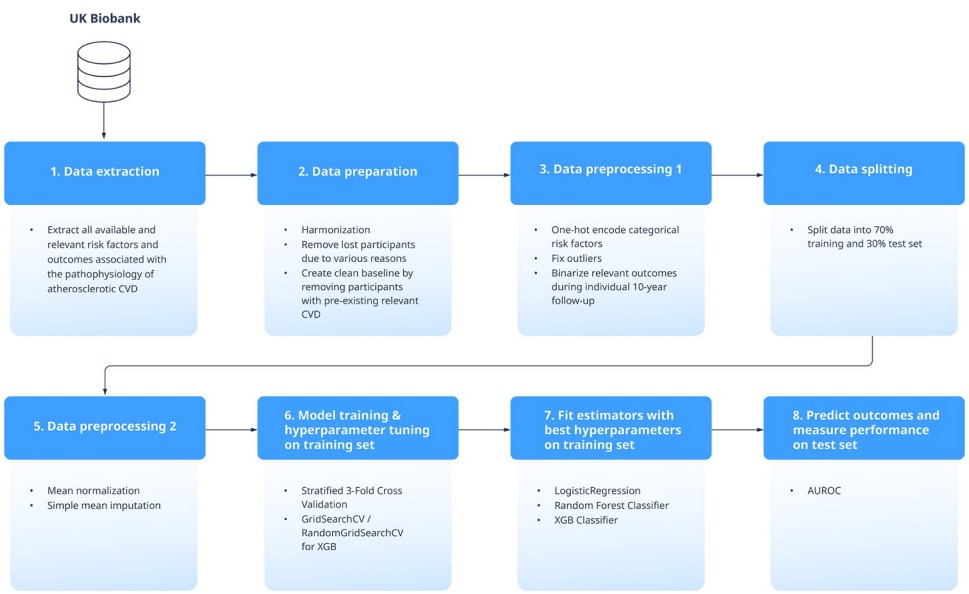

**Fig 1. Overview of experimental setup of proposed approach.**

performance and number of risk factors, with the aim of creating a risk prediction algorithm that is applicable in the clinical context. We used the standard L1 regularization (also known as Lasso) proposed by [57]; it implements a strong penalty on non-zero feature weights of our logistic regression model, resulting in a sparse feature set for prediction.

A logistic regression coefficient value **β** can be interpreted as the expected change in log odds of having the outcome per unit change in the feature $x_{\boldsymbol{\beta}}$. Therefore, increasing the feature by one unit multiplies the odds of having the outcome by $e^{\boldsymbol{\beta}}$. This means that we can interpret the coefficients as feature importance values in the sense that the feature with the smallest coefficient has the least importance on model predictions. Importantly, this holds only true in the context of the parameters contained in the current model. Thus, we re-estimate the model after each feature elimination round.

In each iteration, we re-estimated the logistic regression model on the remaining parameters, and then discarded all parameters that were set to zero by the L1 regularization; finally, we also discarded the parameter with the lowest non-zero absolute value.

As an additional step, we created a ranking of the relative feature importance value of each feature by dividing its absolute coefficient weight by the sum of all absolute coefficient weights.

**Statistical analysis.** To reduce overfitting, we evaluated the classification performance of all our benchmarked algorithms by using 3-fold stratified cross-validation and measuring the Area Under the Receiver Operating Characteristic Curve. For the cross-validation, we used a training set with 325,182 participants to train and derive our standard linear and ML models and then assessed the AUROC performance on the held-out test set with 139,365 participants using 203 risk factors respectively. We reported the AUROC and the 95% confidence intervals (Wilson score intervals) for all models and performed a sensitivity analysis using Shapley Additive Explanations (SHAP values) for the best performing linear model.

**Generalizability.** With 442,620 out of the 502,551 participants in the UK Biobank, the cohort has a high proportion (88.1%) of participants with British White ethnicity. In an effort to estimate a proxy for out-of-sample generalizability, we re-trained the two best models, XGB and logistic regression with L1 regularization, only on Whites and tested their performance on

**Table 1. CVD outcomes statistics according to definition in current study and the comparator study definition by Alaa et al. [29].**

| Statistic measured | Number |
|---|---|
| No. of atherosclerotic CVD outcomes that developed in 10-year follow-up according to definition in current study | 28,561 |
| No. of CVD outcomes that developed in 10-year follow-up according to comparator study definition | 28,242 |
| No. of CVD outcomes after 10-year follow-up that overlap in the current study and comparator study definition | 456,184 out of 464,547 (98%) |
| No. of CVD outcomes identified in the current study but not in comparator studies | 4,341 |
| No. of CVD outcomes included in comporator studies, but not in current study | 4,022 |

a non-White test set. The white-only training set consists of 378,836 participants (81.5%). The non-White test set consists of 85,711 participants (18.5%).

## Results

### Characteristics of the training and test populations

Of 502,551 patients in the UK Biobank, we filtered out 7.6% who already experienced a relevant CVD outcome (during or before baseline) and the participants being lost or who withdrew from the biobank. This resulted in 464,547 participants who met the inclusion criteria. 28,561 (6.1%) of those participants developed at least one of the relevant CVD outcomes during their 10-year follow-up period. We used a common 70% of the data as a training set and 30% as a hold-out test set. Table 1 shows the overlap of our atherosclerotic CVD outcome definition with the CVD outcome definition used in the related work approach by Alaa et al. [29]:

### Prediction accuracy

The resulting prediction accuracy of the benchmarked models is depicted in Table 2. We used both Framingham 10-year CVD risk versions, with and without lipids, as well as QRisk3 as baseline models to assess the performance of predicting someone's 10-year risk of developing an atherosclerotic cardiovascular disease based on a holistic set of risk factors, with a focus on actionable risk factors and outcomes. The best performing model was XGB with an AUROC of 75.73%, only marginally higher than the logistic regression model with L1 regularization (75.44%) and substantially better than the Random Forest model (66.90%).

Fig 2 shows the AUROCs of the best performing models XGB and from logistic regression with L1 regularization, which is the simplest model tested and amongst the top two best performing models. Logistic regression comes with the advantages of being interpretable by providing reasoning for its classifications, and being a simple and robust method [36].

In order to better evaluate the clinical implications and significance of our results, we compared the results of our benchmarked models with our baseline models Framingham and

**Table 2. Performance of all tested classifiers including baseline models.**

| No. | Algorithm Name | AUROC and 95% confidence intervals |
|---|---|---|
| 1 | Extreme Gradient Boosting (XGB) | 0.7573 (0.755–0.7595) |
| 2 | Logistic regression with L1 regularization | 0.7544 (0.7522–0.7567) |
| 3 | QRisk3 | 0.725 (0.7226–0.7273) |
| 4 | Framingham Lipid & BMI | 0.680 (0.6775–0.6824) & 0.681 (0.6788–0.6837) |
| 5 | Random Forest | 0.6690 (0.6666–0.6715) |

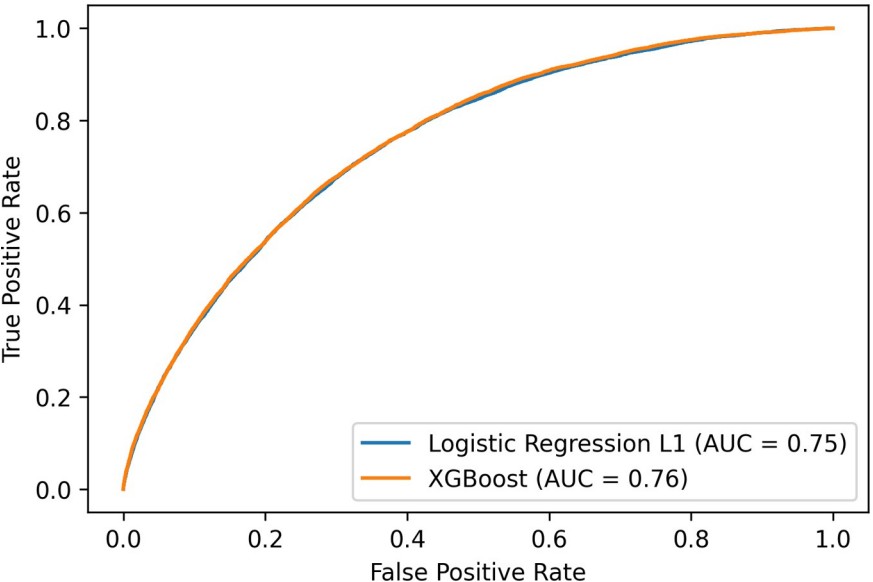

**Fig 2. AUROC of logistic regression with L1 regularization and XGBoost.**

QRisk3. Table 2 shows that both our XGB and logistic regression classifiers achieved superior performance compared to the baseline models. Apart from the Random Forest model, all tested models had a higher AUROC than both baseline Framingham (68.0% and 68.1%) and QRisk3 (72.5%) models.

The difference in AUROC performance of the Framingham score in our experiments in Fig 2 compared to Alaa et al. [29] is explainable by their use of an older UK Biobank version with 40,000 fewer baseline patients with their last available date of participant follow-up being February 17, 2016. The UK Biobank version we used includes biochemistry data which was released May 1, 2019 including cholesterol and additional questionnaires data. Additionally, more diagnosis data was made available over time. These dataset differences may help explain the difference in AUROC.

Figs 3 and 4 show the AUROCs of all baseline models on imputed and unimputed data respectively.

Both Framingham versions perform nearly identically on imputed and unimputed data whereas QRisk3 performs worse on unimputed data.

## Feature elimination vs. predictive performance

Fig 5 shows how the performance of the best logistic regression model depends on the number of risk factors used. Discarding the risk factors stepwise leads to a relatively unchanged and stable model performance until around 170 iterations of feature elimination. This indicates that for predicting an individual's 10-year atherosclerotic CVD risk, many features provide only marginal value and a small subset of features provides substantial informative value. After around 170 iterations, there was a marked decline in model performance associated with further reductions in utilized features.

Table 3 shows in more detail the dependence of the model performance on the number of features. Utilizing only 25 (88%) out of the 203 total risk factors still leads to a reasonable AUROC performance, with a high reduction in utilized features. Compared to the model performance with an AUROC of 75.44% when using all 203 risk factors, the model still achieves 74.15% (95% CI: 0.7392–0.7438) with the 25 most informative risk factors.

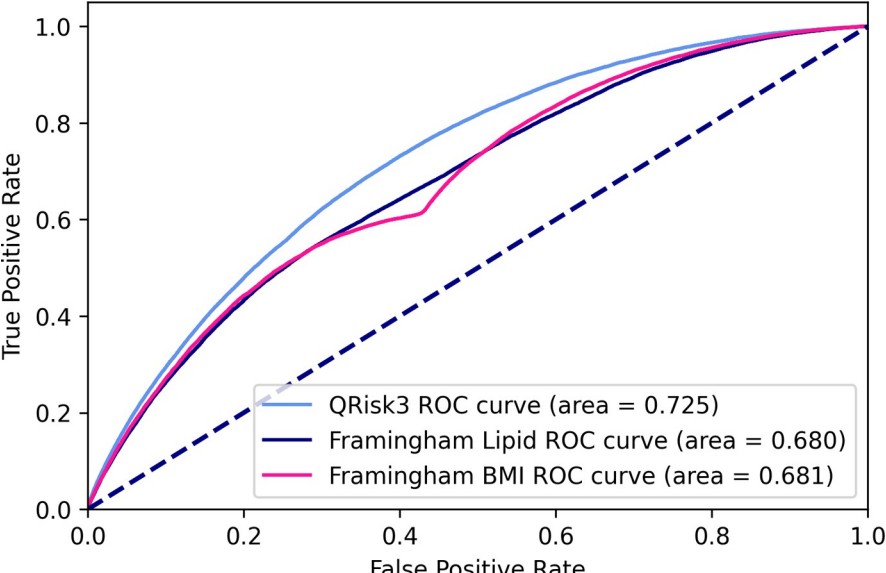

**Fig 3. AUROC curves of baseline models on imputed data.**

We also assessed the performance for fewer features. To reach the same performance as QRisk3 of 72.5% AUROC, 16 features would be necessary. The two most informative features were age and biological sex. To reach a similar performance as Framingham (68.0%), just two features were necessary (68.98%). It is worth noting, however, that both Framingham and QRisk3 were trained and tuned on other datasets and have different CVD definitions and objectives.

## Generalizability of results

We assessed the generalizability of our models by re-training the two previously best performing models only on a White cohort and then testing them on a non-White cohort. Table 4 and

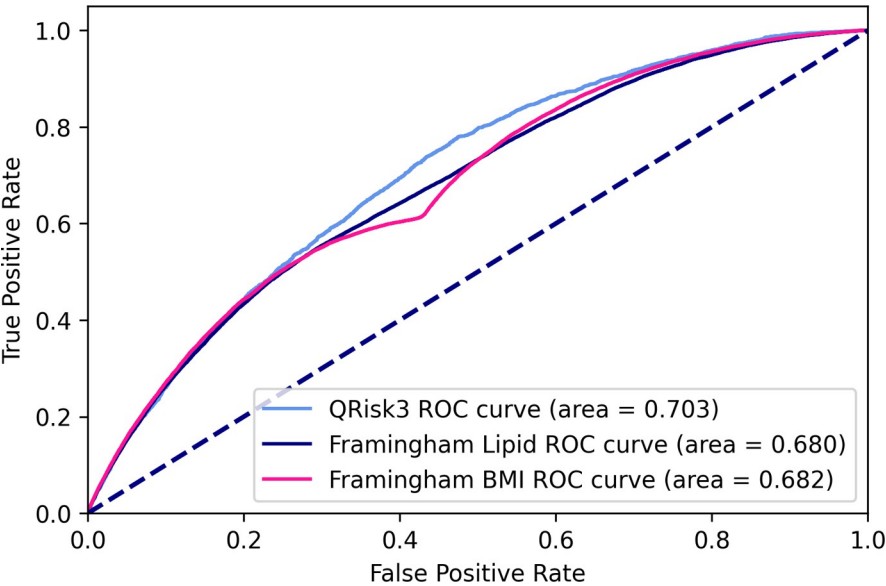

**Fig 4. AUROC curves of baseline models on unimputed data.**

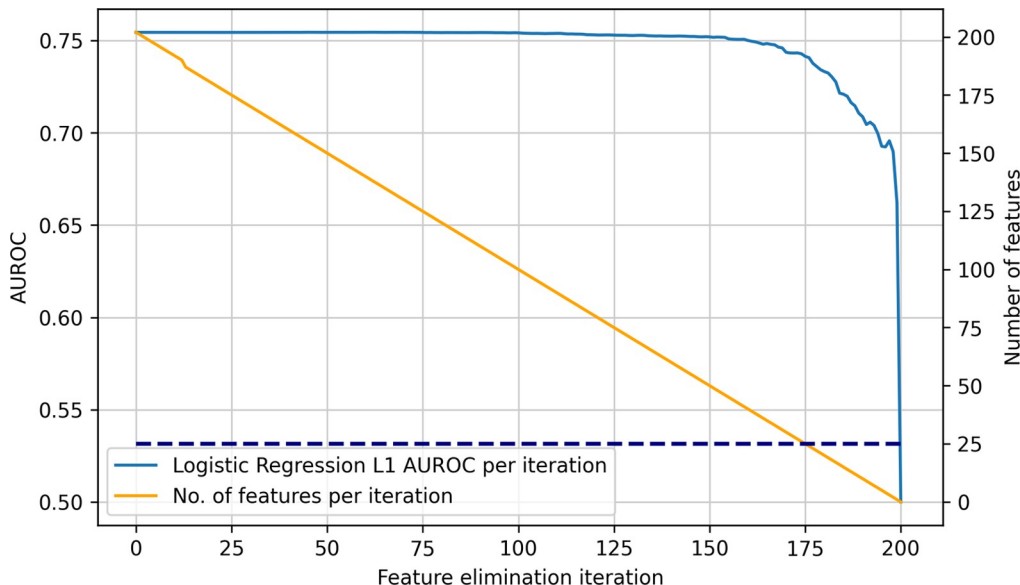

**Fig 5. Performance of best logistic regression model depending on number of features.** AUROC performance of best performing logistic regression model with L1 regularization (continuous blue line) compared to number of features utilized in each iterative feature elimination step (orange line), dotted blue horizontal line showing intersection of 25 features with iterative feature elimination step, allowing for extrapolation to model performance.

Fig 6 show the results for logistic regression and XGB. The logistic regression model has an AUROC of 75.86% in the generalizability experiment, compared with an AUROC of 75.44% in the previous experiment. XGB has an AUROC of 76.26% in the generalizability experiment and 75.73% in the previous experiment. These results show only marginal differences to the results of the previous experiments.

## Predictive ability of individual variables in UK Biobank

Table 5 shows the relative regression feature weights of the 25 most informative risk factors in descending order. A full list is provided in the supplementary materials (S5 Table). Based on our previous manual curation of risk factors and outcomes, we can see that the most informative risk factors are distributed across 5 categories (Table 6), with the lifestyle category contributing the most risk factors. The two most informative features were age and biological sex. We provided a sensitivity analysis using SHAP values of the best performing logistic regression model for all risk factors in the supplementary materials (S1 Fig).

**Table 3. Performance of best logistic regression model depending on number of features.**

| Number of Features | AUROC |
|---|---|
| 203 | 75.44 |
| 40 | 75.01 |
| 25 | 74.15 |
| 20 | 73.32 |
| 17 | 72.76 |
| 10 | 70.88 |
| 2 | 68.98 |

**Table 4. Model performance when trained on Whites and tested on non-Whites.**

| Model | AUROC on generalizability experiment | Previous AUROC results |
|---|---|---|
| Logistic Regression with L1 regularization | 75.86% | 75.44% |
| XGBoost | 76.26% | 75.73% |

## Discussion

Using data gathered from the large longitudinal cohort UK Biobank study, we developed a pipeline to benchmark several classification models for predicting a subject's 10-year absolute risk of developing an atherosclerotic CVD. We used an extensive set of physician curated risk factors and outcomes methodology, employing a holistic view of the subject's current health status rooted in a precision medicine approach. The models were trained and evaluated using data from 464,547 UK Biobank participants, spanning 203 CVD risk factors for each subject. Using a simple logistic regression model with a holistic set of risk factors significantly improved the accuracy of atherosclerotic CVD risk prediction compared to currently available, widely used and recommended models such as Framingham and QRisk3. Both of these existing models rely on a limited set of risk factors and outcomes and do not focus on modifiable lifestyle factors. Further, our best performing logistic regression model utilizes new CVD risk predictors showing high predictive power, namely: social visits, walking pace and overall health rating. The frequency of social visits could be indicative of someone's current mental health status, which has been shown to be a relevant CVD risk factor [58, 59]. These and other non-laboratory risk factors could be collected by means of a questionnaire or passively deduced using data analytics from data sources such as GPS, calendar and sensors [26, 60] from e.g. smartphones, smartwatches and fitness trackers.

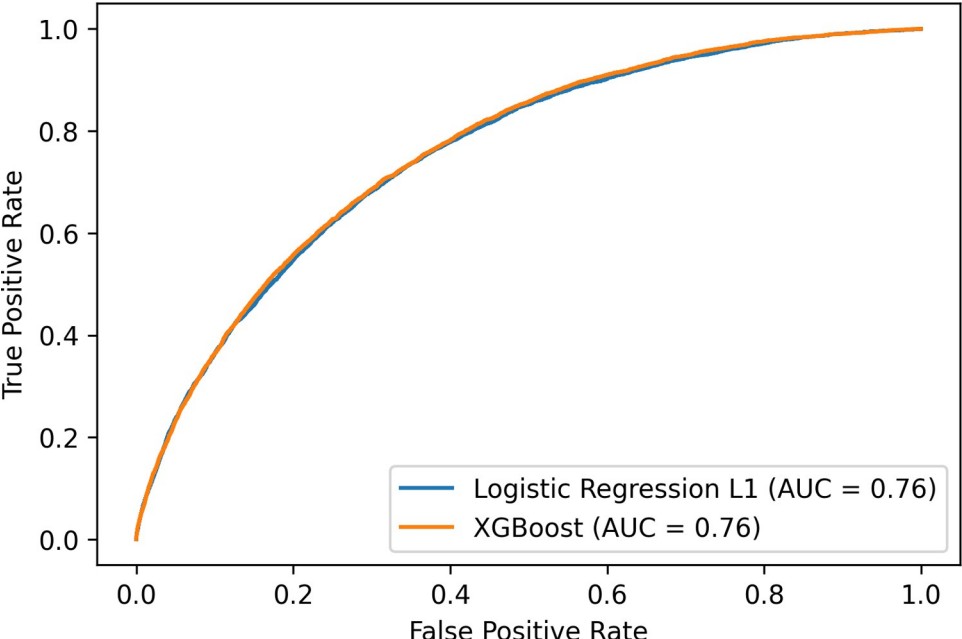

**Fig 6. AUROC of logistic regression with L1 regularization and XGBoost when trained on Whites and tested on non-Whites.**

**Table 5. Relative regression feature weights of 25 most informative risk factors from best logistic regression model.**

| Feature number | Risk factor name | Relative informative value descending |
|---|---|---|
| 1 | Age | 0.0938 |
| 2 | Biological sex | 0.0485 |
| 3 | Systolic blood pressure | 0.0284 |
| 4 | Social visits: About once a week | 0.0277 |
| 5 | Social visits: 2–4 times a week | 0.0273 |
| 6 | Walking pace: Brisk pace | 0.0268 |
| 7 | Total cholesterol HDL ratio | 0.0267 |
| 8 | Total cholesterol | 0.0239 |
| 9 | LDL cholesterol | 0.0235 |
| 10 | Familial CVD | 0.0218 |
| 11 | Social visits: About once a month | 0.0203 |
| 12 | Sleep problems: Not at all | 0.0188 |
| 13 | Alcohol with meals: Yes | 0.0184 |
| 14 | Smoking | 0.0184 |
| 15 | Social visits: Almost daily | 0.0178 |
| 16 | No. of cigarettes daily | 0.0163 |
| 17 | Hypertension | 0.0160 |
| 18 | Walking pace: Steady average pace | 0.0154 |
| 19 | Waist circumference | 0.0150 |
| 20 | Alcohol with meals: It varies | 0.0141 |
| 21 | Social visits: Once every few months | 0.0139 |
| 22 | Overall health rating: Excellent | 0.0134 |
| 23 | Other Heart Arrhythmias | 0.0129 |
| 24 | Overall health rating: Poor | 0.0123 |
| 25 | Sleep problems: Several days | 0.0122 |

Additionally, our best performing models, XGBoost and logistic regression, showed marginal differences when trained and tested on particular sub-populations, which is indicative of good generalizability to other ethnicities.

As there was little performance difference between the best performing models, we primarily discuss the simplest model, logistic regression with L1 regularization. This model has the inherent benefit of offering reasoning for its predictions through analyzing the learned coefficients for every risk factor and having feature selection performed by the L1 regularization.

**Table 6. Categorization of the 25 most informative risk factors into categories from the best logistic regression model.**

| Category | Risk Factors |
|---|---|
| Demographics | Age, Biological sex |
| Biomarkers | Waist circumference, systolic blood pressure, total cholesterol, LDL cholesterol, total cholesterol HDL ratio |
| Comorbidities | Hypertension, sleep problems: not at all, sleep problems: several days, other heart arrhythmias |
| Family History | Familial CVD |
| Lifestyle Factors | Social visits: about once/week, social visits: 2–4 times/week, social visits: about once/month, social visits: almost daily, social visits: once every few months, smoking, no. of cigarettes daily, alcohol with meals: yes, alcohol with meals: it varies, walking pace: steady average pace, walking pace: Brisk pace, overall health rating: excellent, overall health rating: poor |

With L1 regularization, less important risk factors' coefficients are minimized and also set to zero, which then leads to entire removal of these features from the model, and fewer risk factors needed for an accurate prediction.

Using iterative feature elimination, we identified a subset of the 25 most relevant risk factors providing a similar performance compared to using all 203 risk factors. The 25 most relevant risk factors are distributed across five different categories, suggesting that different biological layers contribute to the risk of atherosclerotic CVD. This result confirms that it is insufficient to assess only one biological layer for accurate risk prediction, supporting our initial model development approach [61]. Our approach takes into account multiple biological layers by using multi-omics as well as clinical and lifestyle data with the aim to capture all potential interactions or correlations detected between molecules in different biological layers [22]. Multi-omics data generated for the same set of samples can provide useful insights into the interaction of biological information at multiple layers and thus can help in understanding the mechanisms underlying the complex biological condition of interest.

In our model, the lifestyle category contributed the most risk factors, suggesting that accurate prediction relies upon continuous daily lifestyle data and not just periodic snapshots of clinical data. The causal relationships between the risk factors considered in our model and atherosclerotic CVDs have been demonstrated by other studies [11, 19, 21, 25].

Innovative approaches are needed in order to tackle the increasing prevalence and mortality of CVD-related diseases [2], and the associated healthcare systems' financial burdens. This is particularly true in low and middle income countries where CVD prevalence has also been increasing and is expected to increase as a consequence of an aging and growing population [2]. Our atherosclerotic CVD prediction model has the potential to support healthcare systems by identifying more people at risk earlier and more accurately than currently available models and intervening with personalized behavior change programs. Currently available models, like Framingham and QRisk3, have limited predictive capability for atherosclerotic CVDs as they were not trained on all of them and do not provide actionable results.

There is potential for novel disruptive approaches to affordably improve CVD outcomes. Areas where this may have an impact is in novel approaches to screening, lifestyle coaching and prevention [2]. Screening will become more accessible and widespread by more (near-) medical-grade sensors being integrated into smartphones and smartwatches, enabling continuous monitoring of relevant behavioral CVD risk factors, as well as biomarkers such as heart rate, blood pressure and blood glucose. By gathering a wider spectrum of relevant risk factors for cardiovascular disease automatically and continuously, an ongoing and personalized cardiovascular disease risk prediction could be enabled. Through linking personalized information on an individual's CVD risk with app-based programs for sustained behavioral modification, it may be possible to lower the incidence and mortality of CVDs [62]. Combined with a companion smartphone-based app, an AI or healthcare provider-generated personalized intervention program could be provided and targeted at those people who need it the most.

A system and method gathering personal health data and predicting an individual's atherosclerotic CVD risk is handling sensitive health data (e.g. laboratory values) and must adhere to local regulations and best practices in data transfer, processing and storage to ensure data privacy and security.

Many studies have shown that digital health interventions are cost effective for managing CVD (for a review see [63]). One report found that a community-based prevention program could have a mean return on investment (ROI) on medical cost savings of $5.60 for every $1 spent within a 5 year timeframe by improving physical activity and nutrition and reducing tobacco usage [64]. A review of 11 in-home cardiac rehabilitation programs for the secondary

prevention of CVD found that social support, goal setting, monitoring, credible instructions and literature resources are all effective behavior change techniques to reduce behavioral risk factors for CVD [65].

The improvement achieved by our models might be partially attributed to being trained and assessed on the UK Biobank dataset, whereas the baseline Framingham model was derived from a different population. The population and many of the data sources used in the QRisk3 model are similar, being the general UK population and using their GP, hospital and mortality records. However, our risk model generation approach and QRisk3's approach were designed with different aims and objectives and the modelling strategy was different. For these reasons, direct comparison between the models is limited. Notable differences between the approaches include a more limited set of risk factors included in Framingham and QRisk3's and a focused and wider range of atherosclerotic CVDs included in our approach.

The results from our generalizability sub-analysis indicate that our XGB and logistic regression models might generalize well to other ethnicities and do not overfit to our cohort, however, this needs to be further evaluated with more data from diverse ethnicities.

Our results show that our models have improved performance over the baseline models Framingham and QRisk3 (Table 2). This is because the selection of the appropriate disease modelling approach, classifiers and careful tuning of the model's hyperparameters are crucial steps for realizing the potential benefits of ML. Our pipeline automates some of these steps which makes the tuning and discovery of new disease risk models easily accessible for clinical research. Our prospective cohort modelling approach, which is rooted in precision medicine, is the first to generate an atherosclerotic CVD absolute risk prediction tool based upon a complete definition of atherosclerotic CVD outcomes and a holistic set of risk factors.

## Limitations

The UK Biobank only admitted participants for their initial signup from the ages 40 and up. This might limit the applicability of the risk score for younger populations and further tests with data from younger populations need to be conducted.

There are many missing data values related to the potential risk factors for many participants. Having more unimputed data of relevant CVD risk factors could improve the predictive performance of all our benchmarked classifiers and could also lead to changes in the classifier ranking from Table 2 and relative risk factor importances in Table 5. However, the use of imputed data is highly unlikely to have an impact on our conclusion that a holistic set of risk factors and an exhaustive atherosclerotic CVD outcome definition could improve atherosclerotic and actionable CVD risk prediction.

An additional limitation of our study is that the UK Biobank dataset consists of participants of predominantly (88%) British ethnicity, with an even larger portion having a White background (91%). Therefore, further assessments of the influence of the ethnicity predictor need to be carried out to enable a generalizable tool. Previous work in this area indicates that the development of plaques seems to be independent of ethnicity [21].

A further limitation of this UK-focused dataset is that socio-economic and other environmental factors differ between countries. This is another potential bias that needs to be further evaluated with datasets from other countries with different socio-economic characteristics.

Disease risk prediction models which include subjective non-laboratory risk factors, such as the self-reported health rating and usual walking pace, should be cautiously evaluated to minimize self-reported bias. These risk factors have been found to be good predictors of overall CVD risk in another study using UK Biobank data [29].

## Conclusions

We benchmarked multiple classifiers to predict an individual's 10-year risk of developing an atherosclerotic CVD, using a holistic set of risk factors and a specific definition of atherosclerotic CVDs. Our reduced logistic regression with L1 regularization classifier, a simple and interpretable model, is amongst our best prediction models, includes actionable lifestyle factors, has great predictive power and requires 13 unique features. Our experiments showed that a two feature-questionnaire is as accurate as the Framingham models and a 16 feature-questionnaire is as accurate as QRisk3 for 10-year atherosclerotic CVD risk prediction. Both prediction models, XGBoost and logistic regression, generalize well to non-White people, which might indicate that our models generalize well to other (western) countries. Framingham and QRisk3, which are well established and validated absolute risk prediction models, do not perform as well on predicting individuals' 10-year risk of developing an atherosclerotic CVD. With our logistic regression model, we created a promising new interpretable, actionable and accurate risk prediction tool that could assist individuals and public health in CVD risk reduction.

## Supporting information

**S1 Fig. Shapley Additive Explanations (SHAP value) of each risk factor for the logistic regression model.** This summary plot combines risk factor importance with risk factor effects. It shows the relationship between the value of a risk factor and its impact on the prediction. Risk factors are sorted according to their importance along the y-axis. Each point in the summary plot is a Shapley value for a risk factor and an instance. The position of a Shapley value on the y-axis is determined by the risk factor importance and on the x-axis by the Shapley value. The color represents the value of a risk factor from low to high. Overlapping points are jittered on the y-axis direction, showing the distribution of the Shapley values per risk factor. (TIFF)

**S1 Table. List of all risk factors used in our analysis.** The listed risk factors were summarized into 203 risk factors for the respective UK Biobank participant.
(XLSX)

**S2 Table. List of all outcomes used in our analysis.** The following outcomes were all consolidated into one final binary outcome column indicating if the respective UK Biobank participant did or did not develop one the relevant atherosclerotic CVDs during their individual 10-year follow-up period starting from their individual initial assessment attendance date.
(XLSX)

**S3 Table. Specifications of the python (v3.9.6) libraries and their versions used in this study.**
(PDF)

**S4 Table. List of utilized open-source methods, best parameters and references.**
(PDF)

**S5 Table. Full list of relative informative values for each risk factor for logistic regression model.**
(XLSX)

## Author Contributions

**Conceptualization:** Ajay Kesar, Bernard Leon Stopak, Paul Wicks, Stephen Gilbert.

**Data curation:** Ajay Kesar, Adel Baluch, Omer Barber, Milan Jovanovic.

**Formal analysis:** Ajay Kesar, Daniel Renz.

**Funding acquisition:** Henry Hoffmann, Bernard Leon Stopak, Stephen Gilbert.

**Investigation:** Ajay Kesar.

**Methodology:** Ajay Kesar, Daniel Renz.

**Project administration:** Ajay Kesar.

**Resources:** Ajay Kesar, Henry Hoffmann, Bernard Leon Stopak, Stephen Gilbert.

**Software:** Ajay Kesar, Daniel Renz.

**Supervision:** Henry Hoffmann, Stephen Gilbert.

**Validation:** Ajay Kesar, Daniel Renz.

**Visualization:** Ajay Kesar.

**Writing – original draft:** Ajay Kesar, Daniel Renz, Paul Wicks, Stephen Gilbert.

**Writing – review & editing:** Ajay Kesar, Henry Hoffmann, Daniel Renz, Bernard Leon Stopak, Paul Wicks, Stephen Gilbert.

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
