## [Decision Letter · Decision Letter 0]

19 Dec 2021

PONE-D-21-37349Actionable absolute risk prediction of atherosclerotic cardiovascular disease: a behavior-management approach based on data from 464,547 UK Biobank participantsPLOS ONE

Dear Dr. Kesar,

Thank you for submitting your manuscript to PLOS ONE. After careful consideration, we feel that it has merit but does not fully meet PLOS ONE’s publication criteria as it currently stands. Therefore, we invite you to submit a revised version of the manuscript that addresses the points raised during the review process.

ACADEMIC EDITOR:

Based on the comments from the reviewers and my own observation, I recommend major revisions for the article. It is not mandatory to cite the articles suggested by the reviewers. If the authors feel that the suggested references do not enhance the literature survey they need not cite them.

We look forward to receiving your revised manuscript.

Kind regards,

Thippa Reddy Gadekallu

Academic Editor

PLOS ONE

Journal Requirements:

"All of the authors are or were employees of, contractors for, or hold equity in Ada Health GmbH. AK, AB, OB, HH, MJ, DN, BLS and SG are employees or company directors of Ada Health GmbH and some of the listed authors hold stock options in the company. Ada Health GmbH has received research grant funding from the Bill & Melinda Gates Foundation, Fondation Botnar, the Federal Ministry of Education and Research Germany, the Federal Ministry for Economic Affairs and Energy Germany and the European Union. PW is employed by Wicks Digital Health Ltd, which has received funding from Ada Health, AstraZeneca, Baillie Gifford, Biogen, Bold Health, Camoni, Compass Pathways, Coronna, EIT, Endava, Happify, HealthUnlocked, Inbeeo, Kheiron Medical, Lindus Health, Sano Genetics, Self Care Catalysts, The Learning Corp, The Wellcome Trust, THREAD Research, VeraSci, and Woebot. HH is the topic driver of the AI-based symptom assessment group of the WHO/ITU Focus Group on AI4H (Artificial Intelligence for Health) and SG is a member of the clinical evaluation topic group of the WHO/ITU Focus Group on AI4H.

A related patent application is currently pending with the title “System and method for predicting the risk of a patient to develop an atherosclerotic cardiovascular disease” and application number EP21191089.8."

We note that you received funding from a commercial source: Ada Health GmbH

3. We note that you have a patent relating to material pertinent to this article. Please provide an amended statement of Competing Interests to declare this patent (with details including name and number), along with any other relevant declarations relating to employment, consultancy, patents, products in development or modified products etc. Please confirm that this does not alter your adherence to all PLOS ONE policies on sharing data and materials, as detailed online in our guide for authors http://journals.plos.org/plosone/s/competing-interests by including the following statement: "This does not alter our adherence to  PLOS ONE policies on sharing data and materials.” If there are restrictions on sharing of data and/or materials, please state these. Please note that we cannot proceed with consideration of your article until this information has been declared.

Reviewers' comments:

Reviewer's Responses to Questions

**Comments to the Author**

1. Is the manuscript technically sound, and do the data support the conclusions?

Reviewer #1: Yes

Reviewer #2: Yes

2. Has the statistical analysis been performed appropriately and rigorously? 

Reviewer #1: Yes

Reviewer #2: No

3. Have the authors made all data underlying the findings in their manuscript fully available?

Reviewer #1: Yes

Reviewer #2: Yes

4. Is the manuscript presented in an intelligible fashion and written in standard English?

Reviewer #1: Yes

Reviewer #2: Yes

5. Review Comments to the Author

Reviewer #1: The proposed research discusses a risk prediction approach for diagnosing cardiac arrest risk. It is an interesting research area. However, the following concerns should be addressed.

• The architecture looks very abstract and misses very important details. I recommend authors elaborate design and experimental setup of the proposed approach. The authors have described the materials and methods section, but I recommend including a detailed experimental setup for a better understanding and interpretation of the proposed work.

• A detailed, layered design describing the proposed approach should be included for a better understanding of readers.

• The authors have included 54 references (which occupies a lot of space), which has some unnecessary references which can be removed and essential references such as, https://www.frontiersin.org/articles/10.3389/fpubh.2021.762303/full”, “https://ieeexplore.ieee.org/abstract/document/9170666/” can be referred.

• The results and discussions about how the proposed approach enhances the state of the art is missing. I recommend authors to highlight the contribution of the proposed work separately, along with the limitations of the system.

• The authors have not discussed the security and privacy aspects of the proposed system.

Some more changes are needed:

1. All tables should be symmetrical and should follow a similar formatting style.

2. All the equations should be written using a professional equation editor and should use a similar formatting style and numbering.

3. Check the entire manuscript for grammatical and typo errors.

Reviewer #2: In this manuscript, some machine learning approaches were employed to make a relation among risk and some input factors. Topic is interesting. Please consider the following comments to improve its quality.

-Abstract: please mention results of study in this section.

-Title: I think second part of the title can be reduced and integrated with first part

a behavior-management approach based on data from 464,547 UK Biobank participants

-Introduction: why this study is new and novel. Please mention it in the introduction.

-It is recommended to insert a workflow in the methodology section. Moreover, please describe method briefly in first paragraph of the method.

-I cannot understand why these machine learning approaches were employed.

-I would like to know the selected parameters for running each machine learning approach. It is necessary to change parameters and achieve accuracy result. In fact, a sensitivity analysis should be performed.

6. PLOS authors have the option to publish the peer review history of their article (what does this mean?). If published, this will include your full peer review and any attached files.

Reviewer #1: No

Reviewer #2: No

---

## [Author Response · Author response to Decision Letter 0]

18 Jan 2022

Ajay Kesar

Ada Health GmbH

Karl-Liebknecht-Str. 1

10178 Berlin, Germany

science@ada.com

Dear Mr. Gadekallu,

We thank the editor and the two reviewers for their comments on our manuscript. Below is our revised competing interests statement and responses to each point raised by the academic editor and reviewers. We hope that we satisfyingly addressed them and that the manuscript will be now suited for publication.

Academic editor:

We have modified the file naming to comply with the style requirements and are now fully compliant with the style requirements.

2. Competing Interests Statement:

We added the first and last sentence to our revised competing interests statement, for further clarification: 

”This research was funded by Ada Health GmbH and has been conducted using the UK Biobank under application number 34802.”

And

“This does not alter our adherence to PLOS ONE policies on sharing data and materials.”

Full revised competing interests statement:

“This research was funded by Ada Health GmbH and has been conducted using the UK Biobank under application number 34802. All of the authors are or were employees of, contractors for, or hold equity in Ada Health GmbH. AK, AB, OB, HH, MJ, DN, BLS and SG are employees or company directors of Ada Health GmbH and some of the listed authors hold stock options in the company. Ada Health GmbH has received research grant funding from the Bill & Melinda Gates Foundation, Fondation Botnar, the Federal Ministry of Education and Research Germany, the Federal Ministry for Economic Affairs and Energy Germany and the European Union. PW is employed by Wicks Digital Health Ltd, which has received funding from Ada Health, AstraZeneca, Baillie Gifford, Biogen, Bold Health, Camoni, Compass Pathways, Coronna, EIT, Endava, Happify, HealthUnlocked, Inbeeo, Kheiron Medical, Lindus Health, Sano Genetics, Self Care Catalysts, The Learning Corp, The Wellcome Trust, THREAD Research, VeraSci, and Woebot. HH is the topic driver of the AI-based symptom assessment group of the WHO/ITU Focus Group on AI4H (Artificial Intelligence for Health) and SG is a member of the clinical evaluation topic group of the WHO/ITU Focus Group on AI4H.

A related patent application is currently pending with the title “System and method for predicting the risk of a patient to develop an atherosclerotic cardiovascular disease” and application number EP21191089.8.

This does not alter our adherence to PLOS ONE policies on sharing data and materials.” 

3. Patent Mention in Competing Interests:

We have declared the requested name and number of the pending patent in the competing interests statement and added the last sentence for further clarification: 

“This does not alter our adherence to PLOS ONE policies on sharing data and materials.”.

4. Data Availability

Please find below our revised data availability statement:

“There are restrictions prohibiting the provision of data in this manuscript. The data were obtained from a third party, UK Biobank, upon application. Interested parties can apply for data from UK Biobank directly, at http://www.ukbiobank.ac.uk. UK Biobank will consider data applications from bona fide researchers for health-related research that is in the public interest. By accessing data from UK Biobank, readers will be obtaining it in the same manner as we did.”

Reviewer #1: 

1. The architecture looks very abstract and misses very important details. I recommend authors elaborate design and experimental setup of the proposed approach. The authors have described the materials and methods section, but I recommend including a detailed experimental setup for a better understanding and interpretation of the proposed work. 

A detailed, layered design describing the proposed approach should be included for a better understanding of readers.

Thank you for your feedback. We have included a new figure 1 on page 12 to describe the design and experimental setup of our approach. 

The following sentence was modified for better visibility on page 12: 

“Details on the used Python libraries, methods and parameters are provided in the supplementary data (S3 and S4 Tables)” and this sentence added: “Fig 1 visualizes an overview of all performed steps of our experimental setup.”

2. The authors have included 54 references (which occupies a lot of space), which has some unnecessary references which can be removed and essential references such as, https://www.frontiersin.org/articles/10.3389/fpubh.2021.762303/full”, “https://ieeexplore.ieee.org/abstract/document/9170666/” can be referred.

We thank the reviewer for their suggestions of relevant literature. We can confirm that both studies are relevant as well and are now referenced in our manuscript. We referenced the first study twice and the second study once. As PLOS One is an online journal we understand there is not a strict limit on the number of references, but are happy to follow guidance from the editorial staff.

3. The results and discussions about how the proposed approach enhances the state of the art is missing. I recommend authors to highlight the contribution of the proposed work separately, along with the limitations of the system.

We have taken this suggestion into account and extended our discussion section on page 25 to highlight the contribution of our proposed model more clearly: 

“Our atherosclerotic CVD prediction model has the potential to support healthcare systems by identifying more people at risk earlier and more accurately than currently available models and intervening with personalized behavior change programs. Currently available models, like Framingham and QRisk3, have limited predictive capability for atherosclerotic CVDs as they were not trained on all of them and do not provide actionable results.”

4. The authors have not discussed the security and privacy aspects of the proposed system.

Thank you for highlighting this important missing aspect. We added the following remarks for completeness on page 26:

“A system and method gathering personal health data and predicting an individual's atherosclerotic CVD risk is handling sensitive health data (e.g. laboratory values) and must adhere to local regulations and best practices in data transfer, processing and storage to ensure data privacy and security.”

5. All tables should be symmetrical and should follow a similar formatting style.

All the equations should be written using a professional equation editor and should use a similar formatting style and numbering.

Check the entire manuscript for grammatical and typo errors.

Thank you for your feedback. We refined all table formatting styles to be more consistent. The whole manuscript was double checked by a native English speaker for grammatical and typographical errors.

Reviewer #2: 

1. Abstract: please mention results of study in this section.

We thank the reviewer for their suggestion and have added the results to the abstract on page 1. While doing so, we also noticed a copy and paste error for the confidence intervals of our best performing Logistic Regression model which we have corrected.

2. Title: I think second part of the title can be reduced and integrated with first part

Thank you for your feedback. We shortened the title to “Actionable absolute risk prediction of atherosclerotic cardiovascular disease based on the UK Biobank” on the author page.

3. Introduction: why this study is new and novel. Please mention it in the introduction.

We appreciate the reviewer’s feedback on that matter and have modified and emphasized our unique contributions with a new second to last paragraph in the Introduction section on page 5:

“The aim of this study was to use a large-data ML approach to develop an actionable absolute risk prediction tool which takes into account the holistic health of an individual. Uniquely, we focussed on behavioral risk factors relating to all atherosclerotic CVD outcomes. Our goal was to have a holistic understanding of an individual's current health status, to better quantify their risk of atherosclerotic CVDs, and to provide actionable advice. Our approach is novel in that we employ a highly holistic understanding of an individual’s current health status, to better quantify their risk of all athersclerotic CVDs. By utilizing a comprehensive set of lifestyle factors, we enable the subsequent suggestion of personalized and actionable advice relating to unhealthy risk factors. Instead of using only a limited set of risk factors, we aimed to achieve this by taking multiple biological layers into account, which include: (i) multi-omics data from blood samples (e.g. lipidome and proteome); (ii) family history (e.g. genome), (iii) lifestyle data, (iv) clinical data and (v) environmental data; along with (vi) an extensive set of risk factors and outcomes.”

4. It is recommended to insert a workflow in the methodology section. Moreover, please describe method briefly in first paragraph of the method.

Thank you for your recommendation. We have included a new figure 1 on page 12 to describe the design and experimental setup of our approach in the methodology section. 

The following sentence was added on page 12: “Fig 1 visualizes an overview of all performed steps of our experimental setup.”.

We also added a new brief summary to the methods section on page 6:

 “Baseline data from the UK Biobank was utilized to extract an extensive set of risk factors and outcomes associated with the pathophysiology of atherosclerotic CVDs. A benchmarking pipeline was used to train and evaluate different standard and ML algorithms for the task of 10-year atherosclerotic CVD risk prediction. The performance was measured using AUROC and compared against the baseline models Framingham and QRisk3, which are widely used and recommended models. We evaluated our best performing models further by analysing the most informative features and assessed model generalizability and created a reduced model.”.

5. I cannot understand why these machine learning approaches were employed.

We certainly want to clarify for our readers why we have employed a ML approach and thank you for the opportunity to expand on our rationale in the text. Specifically, we added further clarifications to the method section on page 10:

“Since the introduction of the classic CVD risk prediction methods, the field of supervised machine learning has developed from classical statistics with the sole purpose of maximizing predictive accuracy with modern statistical methods. Therefore, in addition to using standard linear models, we tested the major ML approaches, covering a wide spectrum of the possible ML design space, to evaluate which model type performs best for our task. Based on our initial benchmarking pipeline results, we focused on reporting the results of the initially best performing models: logistic regression, random forest and XGBoost.”

6. I would like to know the selected parameters for running each machine learning approach. It is necessary to change parameters and achieve accuracy result. In fact, a sensitivity analysis should be performed.

Thanks for your feedback. We have added additional information to address your point in the supplementary file S4 Table “List of utilized open-source methods, best parameters and references”, and here we have provided the parameters of the 3 benchmarked methods. For better visibility, we modified the following sentence on page 12: 

“Details on the used Python libraries, methods and parameters are provided in the supplementary data (S3 and S4 Tables).”

We also added the parameters of the other tested methods to the data supplement file S4 Table. 

Additionally, we performed a sensitivity analysis for our best performing Logistic Regression model using Shapley Additive Explanations (SHAP values) and provided the full analysis as a new figure in the supplementary data S1 Figure. 

We added this sentence to the statistical paragraph of the methods section on page 13:

 [...] “and performed a sensitivity analysis using Shapley Additive Explanations (SHAP values) for the best performing linear model” 

and added the following sentences to the manuscript on page 20: 

“We provided a sensitivity analysis using SHAP values of the best performing Logistic Regression model for all risk factors in the supplementary materials (S1 Fig.)” 

and on the last page 35:

“S1 Fig. Shapley Additive Explanations (SHAP value) of each risk factor for the logistic regression model. (PNG) This summary plot combines risk factor importance with risk factor effects. It shows the relationship between the value of a risk factor and its impact on the prediction. Risk factors are sorted according to their importance along the y-axis. Each point in the summary plot is a Shapley value for a risk factor and an instance. The position of a Shapley value on the y-axis is determined by the risk factor importance and on the x-axis by the Shapley value. The color represents the value of a risk factor from low to high. Overlapping points are jittered on the y-axis direction, showing the distribution of the Shapley values per risk factor.”

We hope these modifications satisfyingly increase the quality of our manuscript.

Sincerely on behalf of all authors, 

Ajay Kesar

---

## [Decision Letter · Decision Letter 1]

31 Jan 2022

Actionable absolute risk prediction of atherosclerotic cardiovascular disease based on the UK Biobank

PONE-D-21-37349R1

Dear Dr. Kesar,

We’re pleased to inform you that your manuscript has been judged scientifically suitable for publication and will be formally accepted for publication once it meets all outstanding technical requirements.

Kind regards,

Thippa Reddy Gadekallu

Academic Editor

PLOS ONE

Additional Editor Comments (optional):

Reviewers' comments:

Reviewer's Responses to Questions

**Comments to the Author**

1. If the authors have adequately addressed your comments raised in a previous round of review and you feel that this manuscript is now acceptable for publication, you may indicate that here to bypass the “Comments to the Author” section, enter your conflict of interest statement in the “Confidential to Editor” section, and submit your "Accept" recommendation.

Reviewer #1: All comments have been addressed

Reviewer #2: All comments have been addressed

2. Is the manuscript technically sound, and do the data support the conclusions?

Reviewer #1: Yes

Reviewer #2: Yes

3. Has the statistical analysis been performed appropriately and rigorously? 

Reviewer #1: Yes

Reviewer #2: Yes

4. Have the authors made all data underlying the findings in their manuscript fully available?

Reviewer #1: Yes

Reviewer #2: No

5. Is the manuscript presented in an intelligible fashion and written in standard English?

Reviewer #1: Yes

Reviewer #2: Yes

6. Review Comments to the Author

Reviewer #1: The presented research work should be shared with the research community. The manuscript can be accepted in as it is form.

Reviewer #2: (No Response)

7. PLOS authors have the option to publish the peer review history of their article (what does this mean?). If published, this will include your full peer review and any attached files.

Reviewer #1: **Yes: **Sharnil Pandya

Reviewer #2: No

---

## [Editor Report · Acceptance letter]

4 Feb 2022

PONE-D-21-37349R1 

Actionable absolute risk prediction of atherosclerotic cardiovascular disease based on the UK Biobank 

Dear Dr. Kesar:

I'm pleased to inform you that your manuscript has been deemed suitable for publication in PLOS ONE. Congratulations! Your manuscript is now with our production department. 

Kind regards, 

on behalf of

Dr. Thippa Reddy Gadekallu 

Academic Editor

PLOS ONE